# Post-Hypoxic Myoclonus Status following Out-of-Hospital Cardiac Arrest—Does It still Predict a Poor Outcome? A Retrospective Study

**DOI:** 10.3390/healthcare10010041

**Published:** 2021-12-27

**Authors:** Anne C. Brøchner, Peter Lindholm, Margrethe J. Jensen, Palle Toft, Finn L. Henriksen, Jens F. Lassen, Søren Mikkelsen

**Affiliations:** 1Department of Anaesthesiology and Intensive Care Medicine, Kolding Hospital, 6000 Kolding, Denmark; 2Department of Regional Health Research, University of Southern Denmark, 5000 Odense, Denmark; Soeren.mikkelsen@rsyd.dk; 3The Prehospital Research Unit, University of Southern Denmark, 5000 Odense, Denmark; 4Department of Anaesthesiology and Intensive Care Medicine, Odense University Hospital, 5000 Odense, Denmark; peter.lindholm@rsyd.dk (P.L.); margrethe.jermiin.jensen@rsyd.dk (M.J.J.); palle.toft@rsyd.dk (P.T.); 5Department of Clinical Research, University of Southern Denmark, 5000 Odense, Denmark; finn.lund.henriksen@rsyd.dk (F.L.H.); Jens.flensted.lassen@rsyd.dk (J.F.L.); 6Department of Cardiology, Odense University Hospital, 5000 Odense, Denmark

**Keywords:** out-of-hospital cardiac arrest, prognostication, post-hypoxic myoclonus

## Abstract

In patients with out-of-hospital cardiac arrest (OHCA), the initial prehospital treatment and transfer of patients directly to intervention clinics—bypassing smaller hospitals—have improved outcomes in recent years. Despite the improved treatment strategies, some patients develop myoclonic status following OHCA, and this phenomenon is usually considered an indicator of poor outcome. With this study, we wanted to challenge this perception. The regional prehospital database in Odense in the Region of Southern Denmark was searched for patients with OHCA from the period of 2011–2016. All 900 patients presenting with a diagnosis of OHCA were included in the study. Patients surviving to the hospital and presenting with myoclonic status were followed for up to one year. Only 2 out of 38 patients with myoclonic status and status epilepticus verified by an EEG survived more than one year. Eleven out of 36 patients with myoclonic status but without status epilepticus survived for more than one year. We found no evidence that myoclonic status is an unmistakable sign of poor outcome when not associated with EEG-verified status epilepticus. The conclusion for clinicians involved in post-resuscitation care is that myoclonic status is uncomfortable to witness but does not necessarily indicate that further treatment is futile.

## 1. Introduction

In Denmark, each year, approximately 5400 patients have an out-of-hospital cardiac arrest (OHCA). The Danish 30 day-survival following OHCA is 10.4% [1]. Two-thirds of patients with OHCA that have a return of spontaneous circulation (ROSC) subsequently dies in the hospital due to severe neurological injury [2]. Whereas resuscitation guidelines previously focused on survival rates after cardiac arrest in general, the focus has now changed from mere survival to survival with acceptable cerebral performance.

Some of the early signs of severe neurological harm are the absence of spontaneous breathing, unconsciousness for more than three days despite discontinuation of sedatives, and absence of brainstem reflexes (pupillary reactivity, spontaneous eye positioning and movements, vestibular–ocular reflexes, corneal reflexes, cough, and gag reflexes). Furthermore, status epilepticus and myoclonus status are conventionally considered as indicators of severe neurological injury following cardiac arrest (CA) [3,4,5,6]. The early onset of myoclonic jerks or myoclonic status is considered a sign of serious and acute oxygen deficiency and, thus, a predictor of poor neurologic outcome, even when treated appropriately [7].

A universal definition of myoclonic status does not exist. However, myoclonic status is often defined as repetitive, generalised, focal or multifocal, motor myoclonic jerks involving the face, limbs, or trunk with a duration of more than 30 min [8,9]. Myoclonic status can—but does not always—include pathological patterns in the electroencephalogram (EEG). Myoclonic movements are often seen between 24 and 72 h after CA but can occur at any time following CA. Myoclonic movements are believed to be caused by increased neuronal excitability and, thus, asynchronicity after (in this context: a hypoxic) brain injury. These rhythmical muscular contractions are not symptoms uniquely related to severe neurological damage and neuronal death, as they can also occur in several forms of epilepsy, metabolic dysfunctions, and as part of functional diseases. Contractions may be observed in all kinds of skeletal musculature but are most often seen in the face, in the abdominal muscle, in the lower part of the upper extremities, and in the lower extremities.

Although myoclonic status and status epilepticus do not constitute the same entity, the two conditions are often confused or erroneously considered as the same phenomenon. In patients with myoclonic status, an EEG is thus considered mandatory to eliminate the possibility of status epilepticus [8,9].

Myoclonic jerks, and, when the duration of the jerks is prolonged, myoclonic status, can be unpleasant to witness, not only for the patient’s relatives but also for the healthcare provider. A common approach to myoclonic jerks or myoclonic status in the intensive care unit is thus to treat these clinical findings with several kinds of anticonvulsive medication, as these will attenuate the increased neuronal excitability that is the source of the jerks.

The anticonvulsive medication used does not ameliorate the outcome but merely masks the symptoms. Unfortunately, anticonvulsive medication has side effects such as lethargy and somnolence and, therefore, the use of anticonvulsive medication can blur the assessment of CA patients’ neurological state.

With this study, we wanted to elucidate the incidence of myoclonic status in patients admitted to a tertiary care hospital following resuscitation after OHCA and clarify if myoclonic status can still be regarded as a sign of an unfortunate outcome after OHCA.

## 2. Materials and Methods

We conducted a retrospective cohort study of all patients with OHCA who were treated by the mobile emergency care unit (MECU) in Odense, Denmark between 1 January 2011 and 31 December 2016.

### 2.1. System Setting

In the region of Southern Denmark, the MECU in Odense services a population of 260,000 citizens, approximately 5% of the Danish population. All calls of distress in the Region of Southern Denmark are handled by one emergency medical dispatch centre [10]. The MECU in Odense consists of one rapid-response car, operating all year round. The rapid response unit is manned with a specialist in anaesthesiology [11]. The MECU is dispatched to 26% of all emergency calls that triggers the dispatch of an ambulance. Among these are all calls in which the emergency medical dispatch centre presumes that a case relates to cardiac arrest [10]. After concluding each emergency run, the anaesthesiologist documents the details of the mission in a registry identifying the patient via the patient’s unique civil personal registration system number [12]. The physician registers the MECU response time, the prehospital assigned diagnosis, the treatment administered, as well as procedures performed and the immediate outcome of the patient.

### 2.2. Data Collection

The data source was the MECU database and the in-hospital electronic medical records. We searched the MECU database for patients presenting with a diagnosis of cardiac arrest (ICD-10 Classification System: I46; I46.0; I46.9) [13] within the study period. After identifying all the eligible patients, details of the patient’s admission to the intensive care unit (ICU) were obtained from the in-hospital electronic patient medical records.

All patient records were manually evaluated by three of the authors (ACB, PL, and MJJ). We defined “myoclonic status” as any written statements in the medical records by the treating physician of myoclonic status for more than 30 min.

As an indicator of neurological injury following OHCA, the Cerebral Performance Category Score (CPC) was used. This score categorises the patients into five categories. A CPC score of 1 is defined as normal neurological function and the ability to return to work while CPC 4 or 5 is considered the poorest score with coma and vegetative state or brain death [14].

### 2.3. Inclusion Criteria

Patients were included if any physician-directed treatment was given prehospitally and the patient survived to the hospital.

### 2.4. Exclusion Criteria

Patients were excluded if they were declared dead at the scene.

### 2.5. Variables

For every patient hospital, the following parameters were registered:Age;Sex;Initial cardiac rhythm;Time to ROSC;EEG obtained at the hospital;Any myoclonic convulsions at the ICU.

For patients with myoclonic status: 1-year survival:Cerebral Performance Score.

### 2.6. Statistical Methods

Demographic data are presented as median and quartiles or range (where appropriate). All data were analysed using non-parametric statistics (Chi-square (2 × k tables) and the Kruskal–Wallis test). Differences were considered significant when *p* < 0.05. All data and tables were categorised and prepared using Microsoft Office Excel 2016 (Microsoft Corporation, Redmond, WA, USA). All statistical calculations were performed using STATA 16 (StataCorp, College Station, TX, USA)

### 2.7. Ethical Considerations

The study was approved by the Danish Patient Safety Authority (ref. nos. 3-3013-786/1 and 3-3013-786/23).

## 3. Results

In the six years, 900 patients were included in the initial screening with an overall 30-day survival rate of 18.2%. Of the 900 patients, 484 patients were declared dead at the scene. Of the remaining 416 patients who were all transported to the hospital, 34 patients did not obtain ROSC at any time but were declared dead at arrival to the hospital. Thus, 382 patients were included in the study as having obtained ROSC following OHCA (see Figure 1)

Of the 382 patients included in the study, 307 patients did not exhibit myoclonic status. Thus, the final study population consisted of 75 patients having myoclonic status following OHCA. Within this population, 92 EEGs were obtained in 74 patients. Thirty-eight of these patients with clinical myoclonic status had EEG-verified status epilepticus (see Figure 2).

There was no association between the occurrence of myoclonic status and sex, age, and time to ROSC. The initial cardiac rhythm, however, differed between the groups of patients with myoclonic status and the group of patients without myoclonic status. Twenty-five of 205 patients (12.2%) with shockable rhythm developed myoclonic status, while 47 of 201 patients (23.4%) with non-shockable rhythm developed myoclonic status. This difference was significant (*p* < 0.004). In 10 patients, the initial cardiac rhythm was not recorded (see Table 1 for patient characteristics)

### Survival in OHCA Patients with Myoclonic Status

Of the 74 patients with myoclonic status with a recording of an EEG, 13 patients survived more than one year. EEG-verified status epilepticus was associated with poorer outcomes, as only 2 of the 38 patients survived more than one year and with CPC 3 and 4, while 11 patients of the 36 patients with myoclonic status—but not EEG-verified status epilepticus—survived for more than one year after the OHCA (*p* < 0.005).

Of the 11 patients surviving for more than one year exhibiting myoclonic status but not EEG-verified status epilepticus, 8 patients displayed a CPC score of two or better. See Table 2.

## 4. Discussion

We found that 19.6% of patients surviving OHCA who were subsequently admitted to a university hospital suffered from myoclonic status after admission. Of these, one patient in six survived for more than one year; one patient in ten had a CPC score of 1. Patients in which myoclonic status was accompanied by EEG-verified status epilepticus had a survival rate of only 5.6%.

### 4.1. Outcome

A formal study correlating the caregivers’ perception of futility with any presence of myoclonic status has not been carried out. However, the normal practice during the study period was to consider the presence of myoclonic status when evaluating if further treatment would be futile. This practice has been supported by the available literature [3,4,5,6,7,16].

However, another study investigating a select group of patients, namely young drug users with myoclonic status after cardiac arrest, reported surprisingly good outcomes [17]. In our study, performed in a population of patients with all-cause OHCA, we found that in patients with myoclonic status without concomitant EEG-verified status epilepticus, 11 patients survived to discharge and were alive for more than one year following the OHCA. Of these 11 patients, 7 were discharged with a CPC score of 1, while 1 patient was discharged from the hospital with a CPC score of 2. This indicates that more than 10% of patients with myoclonic status recover to an acceptable cerebral state. In this context, 10% is well above the definition of futility that classically is a “one in a hundred chance” of success [18].

We believe that there may be various explanations for the favourable long-term outcome that we have found in our study:

### 4.2. The Prehospital Setting

In all of the catchment area, patients are treated by a prehospital service with specialists in anaesthesiology that can make critical medical decisions at the scene and have the competencies to administer a medical treatment equalling the initial treatment offered within the hospital. Furthermore, by alerting the coronary intervention department and subsequently accompanying the patients directly to the catheterisation bench, immediate percutaneous cardiac intervention may be carried out in these patients. In addition to the professional stringency, assistance from non-professional bystanders may contribute. In Denmark, education in CPR is mandatory when acquiring a driver’s license. Furthermore, widespread introduction of first aid courses in bigger workplaces, various Heart Runner programs, and an increased amount of Automatic Extern Defibrillators have been implemented in the region.

### 4.3. The in-Hospital Setting

All patients with an acute need for revascularisation may be treated immediately as invasive cardiology is accessible around the clock. The hospital endorses cardiac arrest protocols that routinely address all possible causes of cardiac arrest. Post-cardiac arrest procedures include inclusive neurological assessment and cerebral computerised tomography. The intensive care treatment following cardiac arrest is performed in highly specialised intensive care units manned with specialists in intensive care medicine and specialists in thoracic anaesthesiology on call at all times.

### 4.4. OHCA, Then and Now

For many years, myoclonic status has been considered an unambiguous sign of poor outcome. However, we believe that this perception relates to previous prehospital and in-hospital set-ups where many of the treating modalities available now were not present, among many:

Increasing rates of bystander CPR [19]; the widespread prevalence of Automatic Extern Defibrillators accessible to laypersons [20]; targeted temperature management [21,22]; the importance of rapid cardiological intervention [23].

Overall, an increasing quality of pericardiac arrest treatment has thus been made available to the public, potentially leading the way for a changing of the perception of myoclonic status in patients resuscitated after cardiac arrest.

### 4.5. Limitations

This is a single-centre study and addresses patients cared for by one organisation only.

### 4.6. Strengths

The strength of the present study is the low number of patients lost to follow-up. This is in particular due to the Danish Civil Registration System, in which each resident is assigned an individual identification number [12].

## 5. Conclusions

Although myoclonic status is considered an indication of a poor prognosis among clinicians treating patients with cardiac arrest, our study reveals that over 10% of patients with myoclonic status recover and are dismissed from the hospital with a CPC of 1 to 2. It is important to state that patients with myoclonic status and concomitant status epilepticus still have a very poor survival rate and, thus, these two groups are two entities and must be separated when the outcome is addressed. In concordance with other studies, we emphasise that the emergence of myoclonic events should not be applied as an early prognosticator [24]. As such we believe that myoclonic status should be considered as a sign of cerebral irritability—not necessarily permanent damage—and thus remind the physician that the factors that govern survival following OHCA are complex and that excellent recovery is a combination of many interventions. Should myoclonic status appear, EEG should be monitored and the myoclonic status should be treated with levetiracetam or sodium valproate in addition to sedative drugs. However, as post-hypoxic myoclonus status is not necessarily an unambiguous sign of poor outcome, entering the presence of myoclonic status into prognostication is not advisable for at least the first three days following cardiac arrest [9]. There is a need for the development of more robust tools of prognostication.

## Figures and Tables

**Figure 1 healthcare-10-00041-f001:**
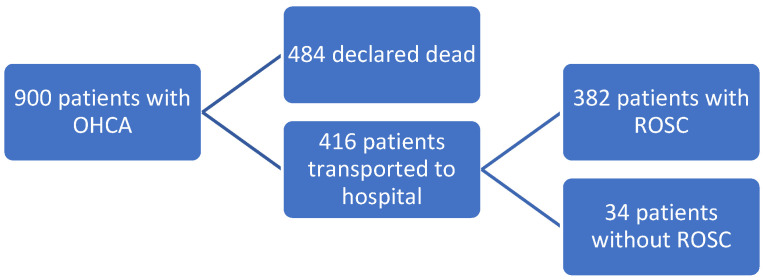
Flow chart showing the investigated patients.

**Figure 2 healthcare-10-00041-f002:**
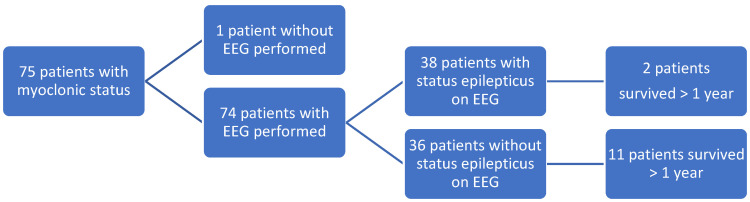
Flow chart showing patients with myoclonic status.

**Table 1 healthcare-10-00041-t001:** Characteristics of the patients. (* Values concerning coronary angiography and revascularisation of “All patients admitted to hospital with ROSC” are quoted from [15].)

Variable	Patients w/ Myoclonic Status and Status Epilepticus	Patients w/ Myoclonic Status	All Patients Admitted to Hospital with ROSC
Shockable Rhythm	14 (36.8%)	25 (33.3%)	205 (49.3%)
Asystole	19 (50%)	37 (49.3%)	127 (30.5%)
PEA	3 (7.9%)	10 (13.3%)	74 (17.8%)
Rhythm Missing	2 (5.3%)	3 (4%)	10 (2.4%)
Male/Female	25/13 (65.8%/34.2%)	53/22 (71.0%/29.0%)	256/126 (67%/33%)
Age (years) (Median (Range))	72 (21–84)	67 (13–91)	66 (1–99)
Time to ROSC or ECMO (min) (Median (Range))	20 (3–35)	18 (3–86)	18 (2–86)
Coronary angiography	18 (47.4%)	39 (52.0%)	85.6% *
Revascularisation	7 (18.4%)	16 (21.3%)	54.8 % *

**Table 2 healthcare-10-00041-t002:** Cerebral Performance Score after one year in patients who presented with myoclonic status but did not display status epilepticus on electroencephalography.

CPC Score	CPC 1	CPC 2	CPC 3	CPC 4	CPC 5
Survivors Discharged	7	1	2	1	n/a

## Data Availability

The datasets used and/or analysed during the current study are available in anonymised form from the corresponding author upon reasonable request.

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
