# Peer review of "Post-Hypoxic Myoclonus Status following Out-of-Hospital Cardiac Arrest—Does It still Predict a Poor Outcome? A Retrospective Study"

_healthcare, 2021, doi:10.3390/healthcare10010041_

Round 1

Reviewer 1 Report

The design and the whole presentation of the study is very professional. The data and the results are well documented and interpreted. Just two questions have arisen in my opinion, I would like the authors to answer/to complete:

  1. would you like provide more data about the myoclonic group, possible differencies to non-myoclonic patients - e.g., clinical diagnoses, left ventricular ejection fraction, ...any clinical, laboratory or therapeutical characteristic you consider important; (please change the title of table 1, the data are just partially demographic)
  2. beyond not to give up therapy,  do you reccomend any specific management of myoclonic cases?

Reviewer 2 Report

Thanks for the oppurtunity to review this manuscript.

Authors did great job in trying to evaluate prognosis of patients with myoclonic status post OHCA.

  • Multiple confounding factors can lead to improved prognosis of the patients with myoclonic status. Without accounting for these confounding factors we cannot come to conclusion that myoclonic status doesnt have poor prognosis, especially when numbers were very small.
  • Authors did mention about some confounding factors - Do you have data on: what kind of arrest (shockable/unshockable), any coronary intervention performed, comorbidities prior to OHCA between status epilepticus and myoclonic status patients or myoclonic status patients and patients without myoclonic status? 
  • 11 patients out of 36 patients with myoclonic status survived one year: 30% survival; Please include whats survival in patients without myoclonic status? And again would be better to understand confounding factors between these 2 groups. 
